# Imbalance Fault Detection of Marine Current Turbine Based on GLRT Detector

**DOI:** 10.3390/s25030874

**Published:** 2025-01-31

**Authors:** Milu Zhang, Jutao Chen, Liu Yang, Christophe Claramunt

**Affiliations:** 1Department of Mechanical and Electrical, Shanghai Maritime University, No.1550 Haigang Avenue, Shanghai 201306, China; jtchen@shmtu.edu.cn (J.C.); liuyang@shmtu.edu.cn (L.Y.); claramunt@ecole-navale.fr (C.C.); 2Naval Academy Research Institute, 29240 Brest, France

**Keywords:** imbalance fault, underwater working environment, GLRT detector, MPM

## Abstract

Marine Current Turbines (MCTs) play a critical role in converting the kinetic energy of water into electricity. However, due to the influence of marine organisms, marine current equipment often experiences imbalance faults. Additionally, affected by the underwater environment, the fault characteristics are submerged in disturbances such as waves and turbulence. Against the background of the above problems, this article proposes a fault detection strategy based on a Generalized Likelihood Ratio Test (GLRT) detector. Firstly, a simulation model of the MCT system is established to obtain prior knowledge. Then, combining the Matrix Pencil Method (MPM) for calculating instantaneous frequency, imbalance fault metrics are selected based on the proposed GLRT detector. At the end, the marine current turbine experimental platform is established, which can simulate imbalanced faults and environmental disturbances, helping to verify the effectiveness of the proposed strategy. The experimental results indicate that the proposed strategy can detect imbalanced faults in complex underwater environments. Imbalance faults are the main manifestation of blade attachments. Thus, it is very meaningful to accomplish fault detection in order to maintain the working order of the MCT system.

## 1. Introduction

With the intensification of energy crises and global environmental pollution, marine current energy has attracted widespread attention [1]. This energy is a kind of efficient, high-quality, clean energy [2]. Marine current turbines play a critical role in converting the kinetic energy of water into electricity [3]. However, MCT typically operates in harsh underwater environments where the effects of saltwater corrosion, attached biomass, random waves and turbulence are significant and inevitable [4]. The growth of marine organisms, the attachment of pollutants, and the corrosion of moving parts often lead to blade imbalance [5]. Factors such as low underwater visibility, random waves and turbulence make it difficult to detect faults [6]. To ensure the safety and reliability of MCT systems, research on imbalance fault detection is crucial [7].

There are multiple ways to detect imbalance fault. Video images of the blade are analyzed in [8], and the VMD method is used to detection biofouling in MCT. In [9], the impact of attaching organisms’ growth on imbalance faults is analyzed. In [10], the method of image recognition for attaching organisms is summarized. However, influenced by the water quality or underwater light, very few photos can be used to detect the blades [11]. Shaft torque signals and vibration signals are obtained for condition monitoring in [12,13]. Frequency domain analysis is used for identifying the characteristic frequencies of imbalance faults. However, the acquisition of these signals requires the support of underwater sensors, which increases maintenance costs.

Detection based on stator voltage or current signals is a non-invasive technique. It is more suitable for underwater applications. The stator voltage signal of the generator is obtained in [14] and [15]. Power Spectral Density (PSD) is directly used to detect faults from the current signal. However, because of the changes in seawater velocity, the imbalance fault characteristics of voltage in different periods expressed different frequency features. Time-frequency analysis methods are suitable under dynamic operating conditions. Short Time Fourier Transform (STFT) is a conventional method for time frequency analysis [16]. However, the frequency resolution of STFT depends on the length of the truncated signal. The resolution of STFT is relatively low under strong interference conditions. Hilbert Transform (HT) shows good performance for estimating the instantaneous frequency and amplitude [17]. However, this method shows serious end effects. In order to fix the above problems, a more precise and robust method for estimating the instantaneous frequency is required. MPM can work efficiently with data of a short length and is robust against interference [18]. It is a good candidate for our application.

Compared to the amplitude of the fundamental frequency, the amplitude of harmonic components caused by imbalanced faults is much smaller [19]. Therefore, fault features may be buried in the fundamental frequency. The solutions mainly focus on two aspects: reducing interference and enhancing feature signals. Empirical mode decomposition is used as a filter to reduce interference for MCT [7]. However, the fault features have also been partially filtered out. Order spectrum analysis transforms equal time interval sampling into equal angle interval sampling to solve the problems of frequency overlap and mixing [20]. However, it requires the precise calculation of rotational speed values, or the addition of additional speed sensors. Synchronous sampling is a potential candidate that can be used to transform the variable fault characteristic frequency into a constant [21]. Furthermore, data normalizations, a method of synchronous sampling [7] designed using prior knowledge for devices in the ocean, is more suitable. In the end, to highlight the amplitude of fault degree, the boundary baseline between normal and faulty must be clearly defined. A detector based on GLRT is designed for the baseline to ensure low false positive and false negative rates [22]. Data normalization and the GLRT detector enhance fault features in two ways.

To attack these problems, this article proposed a strategy consisting of three parts: Instantaneous frequency calculation based on MPM, the GLRT Detector and data normalization. This paper is organized as follows: Section 2 introduces the imbalance fault model. In Section 3, the proposed imbalance fault detection method is presented. In Section 4, the experimental results are given based on an MCT test platform. At the end, a conclusion is drawn.

## 2. Imbalance Fault Model

The MCT system can be described using the following model to obtain prior knowledge. The simplified mathematical model of an MCT system is shown in Figure 1, and it consists of two parts—a turbine and a generator. The turbine converts the kinetic energy of water flow into mechanical energy. The generator converts mechanical energy into electrical energy.

The hydrodynamic torque of the turbine is given as:(1)$$T_{m}=\frac{\rho C_{p}AV_{current}{\,}^{3}}{2\omega_{m}}$$
where ρ is water density, $C_{p}$ is the power coefficient, $A=\pi R_{a}^{2}$ is the swept area, $R_{a}$ is the impeller radius, $V_{current}$ is the incoming flow velocity, and $\omega_{m}$ is the mechanical speed. The tip speed ratio is $\lambda =R_{a}\omega_{m}/V_{current}$. In an ideal environment, $T_{m}$ and $\omega_{m}$ are constant. The formula of motion equation is expressed as:(2)$$J_{m}\frac{d\omega_{m}}{dt}=T_{m}-T_{e}$$
where $J_{m}$ is the moment of inertia and $T_{e}$ is the electromagnetic torque. When the generator is running stably, $d\omega_{m}/dt=0,$$T_{e}=T_{m}$.

### 2.1. Influence of Wave and Turbulence

Affected by noise and interference, the hydrodynamic torque of the turbine can be expressed as(3)$$T_{r}=T_{m}+T_{t}$$
where $T_{t}$ is the torque deviation caused by wave and turbulence. The mechanical speed
$\omega_{r}=\omega_{m}+\omega_{t}$, $\omega_{t}$ is the mechanical speed deviation caused by wave and turbulence. The mechanical speed $\omega_{r}$ can be written as(4)$$\omega_{r}=2\pi (f_{m}+f_{t})$$
where $f_{m}$ is the shaft rotation frequency, and $f_{t}$ is deviation of the shaft rotation frequency. The hydrodynamic torque and mechanical speed of MCT change because of the changing seawater velocity.

### 2.2. Influence of Blade Imbalance on the Stator Current

Affected by blade imbalance faults, the mechanical torque of turbine $T_{n}$ is given by(5)$$T_{n}=T_{r}+T_{im}=T_{m}+T_{t}+T_{im}$$
where $T_{im}$ is the imbalance torque, which can be described as(6)$$T_{im}(t)=\left(mg-\rho gV\right)r_{u}\sin\left(\omega_{r}t\right)$$
where m is the attachment mass, g is the gravitational acceleration, V is the volume of the attachment, and $r_{u}$
is the distance to the center of the attachment. Taking $\left|T_{t}+T_{im}\right|\ll \left|T_{m}\right|$ into account, the electromagnetic torque $T_{e}\approx T_{m}$. In this case, the mechanical speed $\omega_{n}$ can be given as(7)$$\begin{array}{ll} \omega_{n} & =\int_{t_{0}}^{t}\frac{T_{n}-T_{e}}{J_{m}}d\tau \approx \int_{t_{0}}^{t}\frac{T_{n}-T_{m}}{J_{m}}d\tau =\int_{t_{0}}^{t}\frac{T_{im}+T_{t}}{J_{m}}d\tau \\ & =\frac{\rho gV-mg}{J_{m}\omega_{r}}r_{u}\cos(\omega_{r}t)\omega_{t}+\omega_{m}\\ & =\Delta \omega_{r}+\omega_{t}+\omega_{m}=\Delta \omega_{r}+\omega_{r} \end{array}$$
where $\Delta \omega_{r}$ is changes caused by the imbalance fault. The stator voltage of the Permanent Magnet Synchronous Generator (PMSG) can be expressed as(8)$$u_{s}\left(t\right)=U\left(t\right)\cos(\int_{t_{0}}^{t}p\omega_{n}d\tau )=U\left(t\right)\cos(p\omega_{r}t+p\Delta \omega_{r}t+\vartheta )$$
where $U\left(t\right)=C_{n}\omega_{n}$ is the amplitude of the stator voltage and $C_{n}$ is the generator structural constants. ϑ is the initial angle, and p is the number of pole pairs. Using (7) and (8), the instantaneous frequency of the stator voltage can be obtained as(9)$$\begin{array}{ll} f(t) & =\frac{1}{2\pi }\frac{\partial }{\partial t}\int_{t_{0}}^{t}p\omega_{n}d\tau =\frac{1}{2\pi }p(\omega_{m}+\omega_{t}+\Delta \omega_{r})\\ & =p(f_{m}+f_{t})+B\cos[2\pi (f_{m}+f_{t})t] \end{array}$$
where $B\cos[2\pi (f_{m}+f_{t})t]$ is the imbalance fault component. The amplitude $U\left(t\right)$ and frequency f(t) of the voltage vary with the speed of rotation
$\omega_{n}$.

### 2.3. Simulation Results

A simulation model is established according to the simplified mathematical model in Figure 1. The detailed parameters of the MCT system are given in Table 1, including the turbine and generator. The specific impeller parameters in the turbine simulation are given in Table 2. The power coefficient $C_{p}$ is calculated based on blade element theory, and the results are shown in Figure 2a. $C_{p}$ has different values depending on the pitch angle and tip speed ratio. Also, the results for $C_{p}$ are corrected based on the experimental results, which are shown in Figure 2b. This enables the simulation results to reflect the actual situation. During the simulation process, the $C_{p}$ value is obtained using the lookup table method based on Figure 2.

The imbalance fault degree $T_{im}/T_{n}$ is set to 3%. The simulation results are shown in Figure 3. The starting time of the generator is 30 s. Between 30 s and 50 s, the generator runs stably, and is only affected by wave and turbulence. An imbalance fault occurred on the generator after 50 s.

In order to clearly demonstrate the fault characteristics, the voltage envelope and voltage instantaneous frequency are given separately in Figure 4. The voltage waveform after 50 s is analyzed in Figure 4a. It shows the waveform envelopes of a healthy case and a case of imbalance fault. The fault waveform fluctuates based on the healthy waveform. The voltage characteristics are demonstrated in formula (8). The instantaneous frequency comparison for voltage is shown in Figure 4b,c. After 50 s, different performance trends appeared. Affected by the imbalance fault, the instantaneous frequency of the voltage fluctuated based on the health curve. The instantaneous frequency characteristics of the voltage are demonstrated in formula (9).

Via an analysis of the simulation results, it can be concluded that imbalance fault can be identified by detecting the envelope or instantaneous frequency of voltage. But there are two limitations, as follows: (1) in actual MCT imbalance fault detection, there is no healthy voltage curve as a reference for comparison; (2) the amplitude and instantaneous frequency of voltage change frequently, and important features are covered up by the excessive quantity of unrelated features, increasing the imbalance fault distinguish difficulty.

## 3. Fault Detection Method Based on GLRT Detector

In view of the above problems, a fault detection method based on the Generalized Likelihood Ratio Test (GLRT) detector is proposed. Firstly, MPM is used for instantaneous frequency estimation (at this step, the envelope can also be used for imbalance detection, taking instantaneous frequency as an example). Then, a GLRT detector is developed to select an appropriate monitoring variable. At the end, the data normalization method is proposed to solve the problem of the multiple forms of fault feature caused by different flow velocities.

### 3.1. Instantaneous Frequency Estimation Based on MPM

The instantaneous frequency is estimated based on the short-time voltage of length
$N_{s}.$
The short-time signal can be approximately expressed as:(10)$$y(n)=\sum_{k=1}^{M}a_{k}e^{j(\omega_{k}n+\varphi_{k})}+w(n)$$
where $y(n)=u_{s}\left(n/Fs\right)$ denotes the voltage samples, w(n) is a white Gaussian noise, M is the number of complex sinusoids, $\omega_{k}$ is the *kth* frequency, $a_{k}$ and $\varphi_{k}$ are the corresponding amplitude and phase, respectively. There are two steps in MPM, as follows [23]:

(1) Construct a Hankel matrix

The $N_{s}$ samples y(n) (with
$0\leq n\leq N_{s}-1$) allow the construction of the Hankel matrix,(11)$$Y=\left[\begin{array}{cccc} y(0) & y(1) & \cdots & y(L)\\ y(1) & y(2) & \cdots & y(L+1)\\ \vdots & \vdots & \vdots & \vdots \\ y(N_{s}-L-1) & y(N_{s}-L) & \cdots & y(N_{s}-1) \end{array}\right]$$
where L is called the pencil parameter, and $M\leq L\leq N_{s}-M.$
Usually, its value is set between
$N_{s}/4$ and $N_{s}/3$. Ignoring the white Gaussian noise, Y can be rewritten as(12)$$Y=\sum_{k=1}^{M}a_{k}e^{j\varphi_{k}}\left[\begin{array}{cccc} 1 & e^{j\omega_{k}} & \cdots & e^{jL\omega_{k}}\\ e^{j\omega_{k}} & e^{j2\omega_{k}} & \cdots & e^{j(L+1)\omega_{k}}\\ \vdots & \vdots & \vdots & \vdots \\ e^{j(N_{s}-L-1)\omega_{k}} & e^{j(N_{s}-L)\omega_{k}} & \cdots & e^{j(N_{s}-1)\omega_{k}} \end{array}\right]$$

(2) Instantaneous frequency estimation

The matrix pencil is defined as $Y_{2}-\lambda Y_{1},$ where $Y_{1}=Y{\left[I_{L},0_{L\times 1}\right]}^{T}$ and $Y_{2}=Y{\left[0_{L\times 1},I_{L}\right]}^{T}$ are two matrices obtained from the first L columns and last L columns of Y, respectively. $I_{m}$ is the m×m identity matrix; $0_{m\times n}$ is a matrix of dimension m×n with all elements equal to 0; λ is a scalar parameter.$Y_{1}$ and $Y_{2}$ can be written as
(13)$$Y_{1}=Z_{1}RZ_{2}\;Y_{2}=Z_{1}RZ_{0}Z_{2}$$
where $Z_{1}=[z_{1}(\omega_{1}),z_{1}(\omega_{2}),\ldots ,z_{1}(\omega_{M})]$; $Z_{2}={\left[z_{2}(\omega_{1}),z_{2}(\omega_{2}),\ldots ,z_{2}(\omega_{M})\right]}^{T}$;
$$R=\mathrm{diag}[a_{1}e^{j\varphi_{1}},a_{2}e^{j\varphi_{2}},\ldots ,a_{M}e^{j\varphi_{M}}];Z_{0}=\mathrm{diag}[e^{j\omega_{1}},e^{j\omega_{2}},\ldots ,e^{j\omega_{M}}];$$$$z_{1}\left(\omega_{k}\right)={\left[1,e^{j\omega_{k}},\ldots ,e^{j(N_{s}-L-1)\omega_{k}}\right]}^{T};z_{2}\left(\omega_{k}\right)={\left[1,e^{j\omega_{k}},\ldots ,e^{j(L-1)\omega_{k}}\right]}^{T}.$$

The singular-value decomposition of Y results in a diagonal matrix Σ containing the singular values $s_{k}$ of Y arranged in descending order,(14)$$Y=L\Sigma U^{H}$$
where “H” denotes the conjugate transpose. The number of complex sinusoids M can be deduced based on (15)$$\zeta (k)=\sqrt{\frac{s_{1}^{2}+s_{2}^{2}+\ldots +s_{k}^{2}}{s_{1}^{2}+s_{2}^{2}+\ldots +s_{L+1}^{2}}}k=1,2,\ldots ,L+1$$

*M* is determined as $M=\arg\min_{k}\zeta (k)>0.995$. The new matrix $\Sigma^{\prime }$ is made up of the *M* largest singular values, as follows:(16)$$\Sigma^{\prime }={\left[\begin{array}{cccc} s_{1} & 0 & \cdots & 0\\ 0 & s_{2} & \cdots & 0\\ \vdots & \vdots & \ddots & \vdots \\ 0 & 0 & \cdots & s_{M}\\ \, & \, & 0_{(N_{s}-L-M)\times M} & \, \end{array}\right]}_{(N_{s}-L)\times M}$$

The first M columns of U are used to get matrix $U_{s},$ where $U_{s}=U{\left[I_{M},0_{M\times (L+1-M)}\right]}^{T}.$ Let $U_{s}^{↑}=[I_{L},0_{L\times 1}]U_{s}$ and $U_{s}^{↓}=[0_{L\times 1},I_{L}]U_{s}$ be the first L rows and last L rows of $U_{s},$ respectively,(17)$$\left\{\begin{array}{c} Y_{1}=L\Sigma^{\prime }U_{s}^{↑}{\,}^{H}\\ Y_{2}=L\Sigma^{\prime }U_{s}^{↓}{\,}^{H} \end{array}\right.$$

Using (13), the matrix pencil becomes(18)$$Y_{2}-\lambda Y_{1}=Z_{1}R(Z_{0}-\lambda I_{M})Z_{2}$$

Under the condition of $\lambda =e^{j\omega_{k}},$
k=1,2,…,M,
$Y_{2}-\lambda Y_{1}=0$.
Then
$e^{j\omega_{k}}$ can be estimated by solving the following eigenvalue problem:(19)$$G=Y_{1}^{+}Y_{2}$$
where “+” denotes the pseudo inverse operation. G is a matrix whose eigenvalues are the generalized eigenvalues of the matrix pair $[Y_{2},Y_{1}].$
In practice, the noise is always present, in which case total least-squares MPM is used,(20)$$\begin{array}{ll} G & =Y_{1}^{+}Y_{2}=(L\Sigma^{\prime }U_{s}^{↑}{\,}^{H}{)}^{+}(L\Sigma^{\prime }U_{s}^{↓}{\,}^{H})\\ & =(U_{s}^{↑}{\,}^{H}{)}^{+}(\Sigma^{\prime }{)}^{+}L^{+}L\Sigma^{\prime }U_{s}^{↓}{\,}^{H}=(U_{s}^{↑}{\,}^{H}{)}^{+}U_{s}^{↓}{\,}^{H} \end{array}$$

The instantaneous frequency is estimated as(21)fk=Fs2πarctan(Im(λk)Re(λk))
where $\lambda_{k}$ is the *kth* largest eigenvalue of G. Combined with the adjustment of the data length of the moving window, MPM can be used to track the frequency variation with time.

### 3.2. GLRT Detector

We pose the following two hypotheses: the null hypothesis $H_{0}:$—the MCT system is healthy, and the alternative hypothesis $H_{1}:$—imbalance fault occurs in the MCT system. In $f(n)=f(n/F_{s}),$ n is the sampling point n=0,1,…,N−1;, and $F_{s}$ is the sampling rate. This detection problem is described as(22)$$\begin{array}{l} H_{0}:f(n)=p(f_{m}+f_{t})+\xi (n)\\ H_{1}:f(n)=p(f_{m}+f_{t})+Bs(n)+\xi (n) \end{array}$$
where ξ(n) is the additive Gaussian noise, $\xi [n]\sim N_{c}(0,\sigma^{2});$, and s(n) is the fault component,(23)$$s(n)=\cos[2\pi (f_{m}+f_{t})n]$$

Suppose that s(n) is a known quantity and B is an unknown parameter. We outline the following linear model:(24)$$f=\left[\begin{array}{c} f(0)\\ f(1)\\ \vdots \\ f(N-1) \end{array}\right]=\underset{H}{\underbrace{\left[\begin{array}{cc} s(0) & 1\\ s(1) & 1\\ \vdots & \vdots \\ s(N-1) & 1 \end{array}\right]}}\begin{array}{cc} \underset{\theta }{\underbrace{\left[\begin{array}{c} B\\ p(f_{m}+f_{t}) \end{array}\right]}} & +\xi \end{array}$$
where H is the observation matrix, θ is the parameter vector, and ξ is the noise vector. With B=[1 0], the detection problem becomes(25)$$\begin{array}{l} H_{0}:B^{\theta }=0\\ H_{1}:B^{\theta }\neq 0 \end{array}$$

The GLRT-based detector then produces the following:(26)$$\begin{array}{ll} T(f) & =\frac{{\hat{\theta }}_{1}^{T}B^{T}{\left[B{\left(H^{T}H\right)}^{-1}B^{T}\right]}^{-1}B{\hat{\theta }}_{1}}{\sigma^{2}}\\ & =\frac{{\left(\sum_{n=0}^{N-1}\left[f(n)-\bar{f}]s(n\right)\right)}^{2}}{\sigma^{2}\left(\sum_{n=0}^{N-1}s^{2}(n)-\frac{1}{N}{\left(\sum_{n=0}^{N-1}s(n)\right)}^{2}\right)} \end{array}$$
where ${\hat{\theta }}_{1}$ is the maximum likelihood estimator of θ under $H_{1},$${\hat{\theta }}_{1}={\left(H^{T}H\right)}^{-1}H^{T}f;$
$\bar{f}=\frac{1}{N}\sum_{n=0}^{N-1}f(n).$ The GLRT-based detector for the considered problem helps to decide $H_{1}$ if(27)$$[\sum_{n=0}^{N-1}\left(f(n)-\bar{f})s(n\right){]}^{2}>\gamma$$
where γ is a threshold. The value of $f(n)-\bar{f}$ determines whether the detection result exceeds the threshold. If $f(n)-\bar{f}$ is in accordance with s(n), the hypothesis $H_{1}$ is selected. Therefore, $f(n)-\bar{f}$ can be used as the monitoring variable.

### 3.3. Data Normalization

We assume that the original nonstationary stator voltage signal is uniformly sampled with sampling frequency $F_{s}.$ The sampling starting time is $N_{0}/F_{s},$ and the sampling time it takes the blade to rotate Z cycles is $N_{Z}/F_{s}.$ The total number of sampling points in each mechanical cycle is variable because of the changing shaft rotating speed. $N_{1},N_{2},\ldots ,N_{z}$ are the breaking points for each mechanical cycle. From $N_{0}$ to $N_{Z},$ the average number of sampling points in each cycle is $\bar{N}=(N_{Z}-N_{0})/Z.$


At time $n_{t}=(i\bar{N}+n_{c})$(i=0,1,…,Z−1,$0\leq n_{c}\leq \bar{N}-1$), the new instantaneous frequency can be obtained as:


(28)
$$f_{c}(N_{0}+n_{t})=f(N_{i}+\frac{N_{i+1}-N_{i}}{\bar{N}}n_{c})$$


Combining with (22), $f(n)-\bar{f}$ can be rewritten as(29)$$\begin{array}{l} f_{c}(N_{0}+n_{t})-{\bar{f}}_{c}\\ =f_{c}(N_{0}+n_{t})-\frac{1}{\bar{N}}\sum_{n_{t}=i\bar{N}}^{(i+1)\bar{N}-1}f_{c}(N_{0}+n_{t})\\ =\left\{\begin{array}{c} H_{0}:0+b(N_{0}+n_{t})\\ H_{1}:Bs_{c}(N_{0}+n_{t})+b(N_{0}+n_{t}) \end{array}\right. \end{array}$$
where $b(N_{0}+n_{t})$ is the noise component. The period of imbalance fault component $s_{c}(N_{0}+n_{t})$ is constant because of (28). It shows that if $H_{1}$ is true, the statistical mean value of the monitoring variable is approximately equal to $Bs_{c}(N_{0}+n_{t}).$ Therefore, B can be used to recognize the fault degree.

### 3.4. Process of Fault Detection

The complete process is described in Figure 5.

(1)It starts with the non-stationary stator voltage signal of length N. Zero crossing points or extreme points are found to determine the breaking points of N. The data length is the number of samples in an integer number of mechanical cycles.(2)The instantaneous frequency is calculated by MPM. The data begin with $u_{s}\left(n\right),$ and end with $u_{s}\left(n+N_{sl}\right),$ where $N_{sl}$ is the length of the segment including $u_{s}\left(n\right).$(3)The instantaneous frequency f(n) is normalized by (28), and the monitoring variable $f_{c}(N_{0}+n_{t})-{\bar{f}}_{c}$ is calculated by (29). The variable B is used to measure the fault degree.(4)The power spectral density of the monitoring variable $f_{c}(N_{0}+n_{t})-{\bar{f}}_{c}$ is analyzed to detect the fault characteristic frequency.

## 4. Experimental Design and Analysis

### 4.1. Experiment System Setup

A 230 W horizontal-axis direct-drive MCT experimental platform was here designed, as shown in Figure 6. It includes a water tank, an MCT prototype, an adjustable resistor, and a data acquisition system. The volume of the water tank is 45 $m^{3}$. The flow velocity can be adjusted from 0.2 m/s to 1.8 m/s. A 45 kW water pump is used for water circulation. The flow deflector and honeycomb ensure stable flow velocity in the vertical section of the water tank. The detailed parameters of the MCT prototype are given in Table 1. The electrical load adopts pure resistance, with a constant resistance value. The measured voltage signals are collected by the National Instrument data acquisition system with a sampling frequency of 1 kHz.

Figure 7 shows the settings for a healthy case and imbalance fault case in an MCT, respectively. The weight of the attachment is 200 g. Figure 8 gives the stator voltage waveforms at rotor speeds of 7 n/min and 78 n/min under imbalanced fault conditions. At a speed of 7 n/min, the imbalance fault can be clearly observed. The signal intensity here is much smaller than normal. At a speed of 78 n/min, the fault feature is buried in the fundamental wave.

The interference impact is set to add another MCT upstream, as shown in Figure 9. Compared to the curve under stable working conditions, the curve under strong interference working conditions faces more fundamental changes. It shows that MCT is sensitive to the quality of the incoming flow. The wake of the upstream MCT has a greater impact on the downstream MCT.

The original waveform of voltage sampling from the MCT generator is given in Figure 10. The MCT system has an imbalanced fault in the entire waveform. When approaching 14 s, stopping the water pump causes the MCT to stop. The process of MCT generator stopping is marked in the diagram. It can be seen that the envelope of the voltage waveform undergoes certain fluctuations, as described in Formula (8) and in the simulation results in Figure 4a. Figure 11 gives the results of the order analysis. It presents the instantaneous frequency of the voltage and the speed of the generator. Also, the waveform has certain fluctuations, as described in Formula (9) and the simulation results in Figure 4b. The shutdown process of the generator, as shown in Figure 11, is easily identified, but the imbalance fault attributes are not obvious.

### 4.2. Fault Detection Results and Analysis

In Figure 12, the instantaneous frequency is calculated by the STFT, HT, and MPM methods. Table 3 provides a comparison of different time-frequency analysis methods with different fault degrees ($T_{im}/T_{n}=0\%$, 1%, and 3%). The STFT method suffers from low resolution. The HT method has serious end effects. The relative errors of STFT and HT are greater than those of MPM. The time-domain characteristics of instantaneous frequency are highlighted by the MPM method. The detection results show that the value *B* can be used to distinguish different imbalance faults.

For comparison, Figure 13a shows the result without the GLRT detector and data normalization. It is a frequency domain diagram of the stator voltage. The frequency components basically coincide with each other. Influenced by the changing velocity of sea water, imbalance fault features are buried under variable fundamental frequency. Figure 13b shows the frequency domain after data normalization without a GLRT detector, $T_{im}/T_{n}=1\%$, and here, 3% cannot be distinguished. Figure 13c gives the frequency domain diagram of the proposed monitoring variable $f(n)-\bar{f}$ after data normalization. Table 4 provides the result of the proposed fault detection method in different cases. Different from the healthy case, both 1% and 3% imbalance faults show frequency components at 1.92 Hz. The amplitude of the frequency component varies with fault degree. This proposed method can effectively detect the imbalance fault and identify the different fault degrees. The experimental results confirm the effectiveness of the proposed method based on a GLRT detector.

## 5. Conclusions

In this paper, a novel fault detection strategy is proposed to identify blade imbalance faults of a variable-speed direct-drive MCT using MPM, a GLRT detector and data normalization. To realize the fault feature extraction, MPM is used for the instantaneous frequency estimation of short-time stator voltage. For the problem wherein fault features are present in various forms and could be buried in the fundamental frequency and environmental noise, fault metrics are selected based on the GLRT detector and then processed by data normalization. The detection results indicate that the proposed method can accurately determine the fault degree. The limitation of this work is that there was no quantitative analysis performed of the interference caused by waves. This strategy can be used for the long-term monitoring of an MCT system. It has important practical significances, such as ensuring the safe and efficient operation of the MCT system and reducing the maintenance cost.

## Figures and Tables

**Figure 1 sensors-25-00874-f001:**
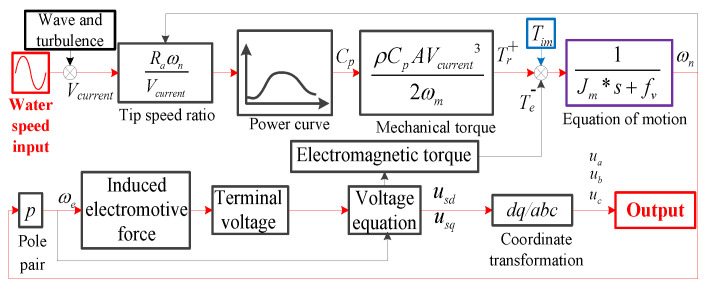
MCT system imbalance fault model.

**Figure 2 sensors-25-00874-f002:**
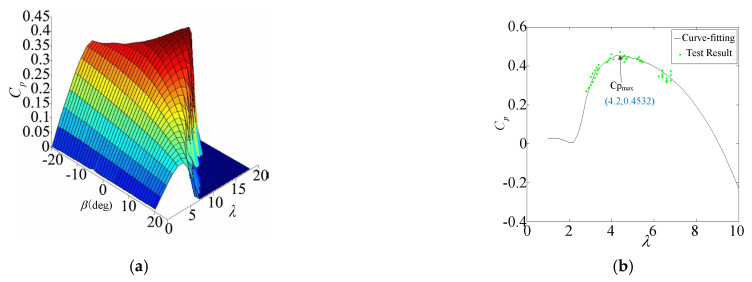
Simulation results of turbine model. (**a**) The relationship between power coefficient, pitch angle, and tip speed ratio. (**b**) Simulation diagram of power coefficient.

**Figure 3 sensors-25-00874-f003:**
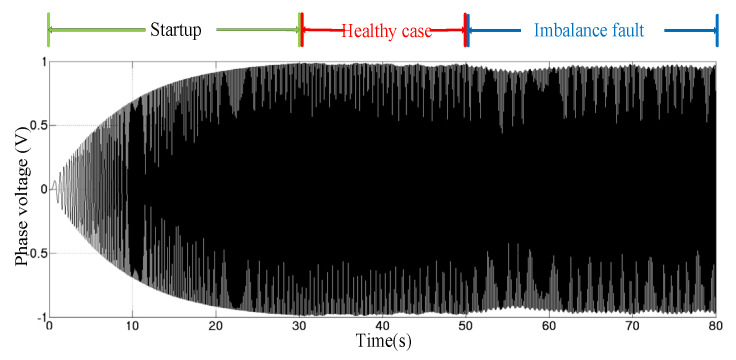
Voltage output of MCT.

**Figure 4 sensors-25-00874-f004:**
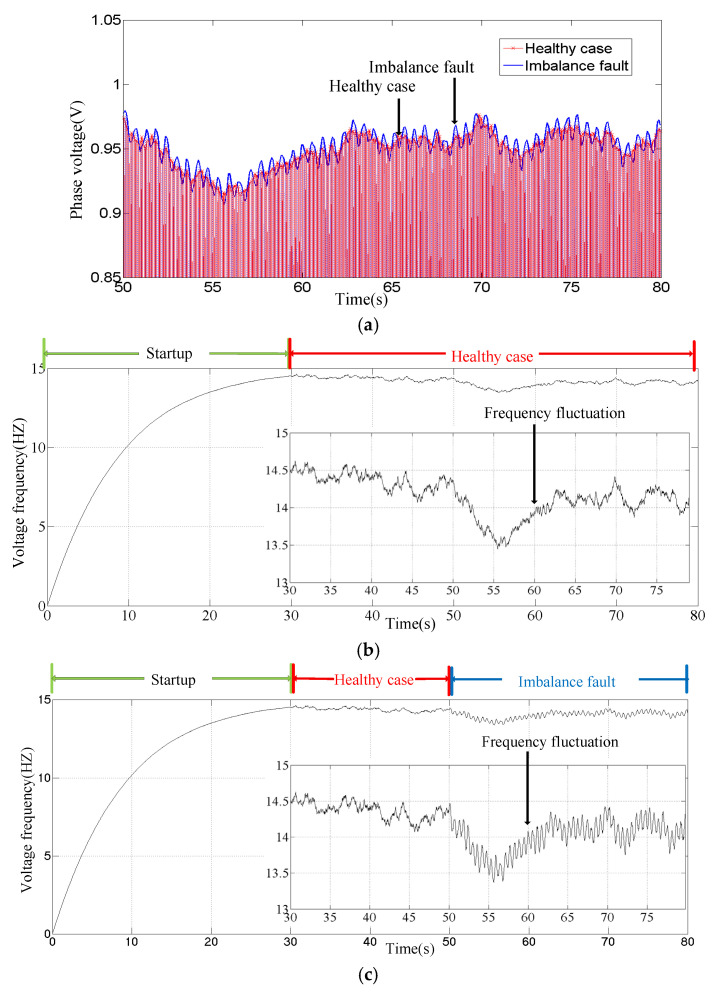
Comparison of the stator voltage under faulty and healthy cases. (**a**) Stator voltage amplitude. (**b**) Stator voltage frequency in healthy case. (**c**) Stator voltage frequency with imbalance fault.

**Figure 5 sensors-25-00874-f005:**
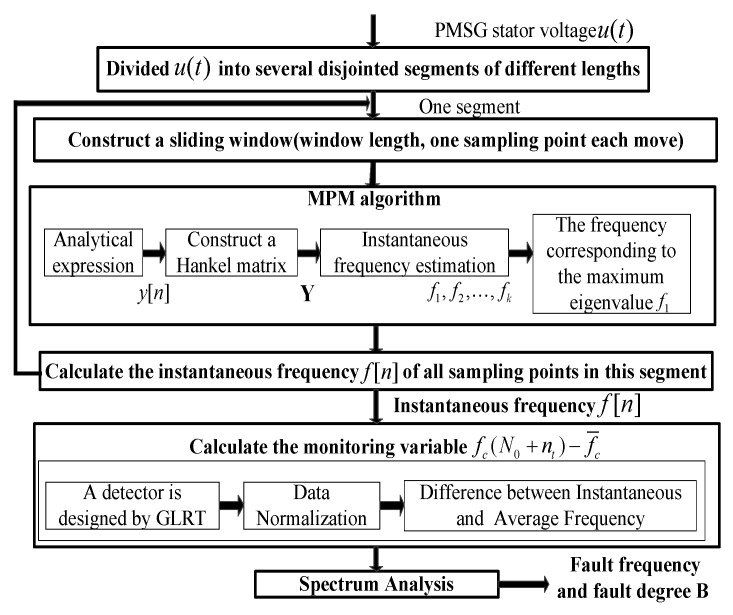
Schematic diagram of the proposed fault detection strategy.

**Figure 6 sensors-25-00874-f006:**
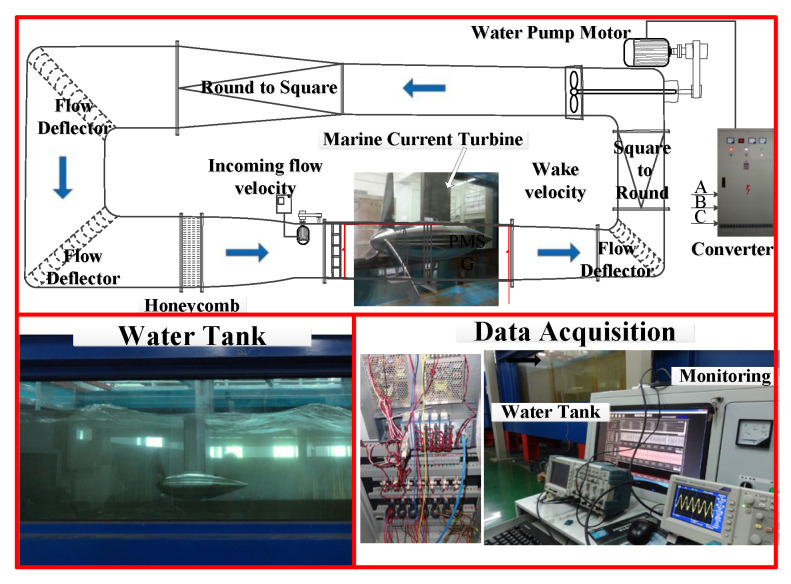
MCT test platform.

**Figure 7 sensors-25-00874-f007:**
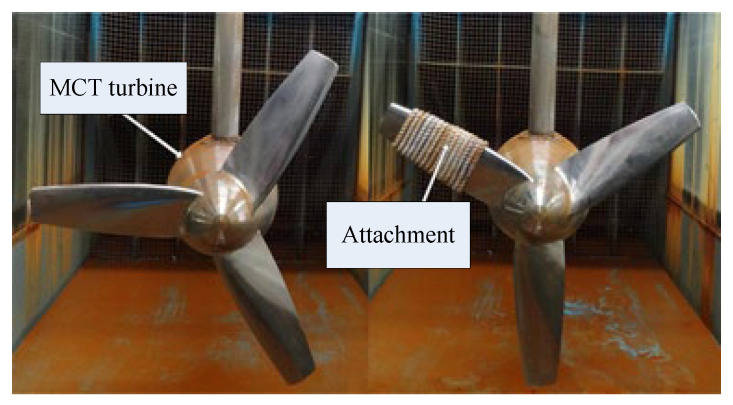
Imbalance fault settings.

**Figure 8 sensors-25-00874-f008:**
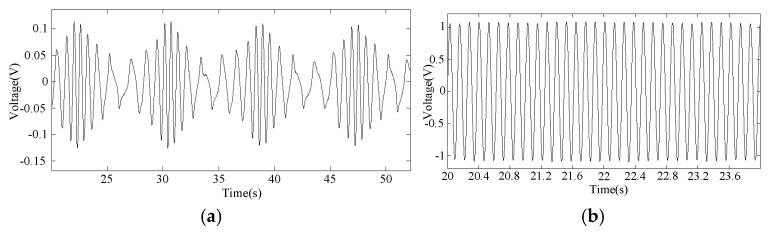
Stator voltage affected by imbalanced faults at different speeds. (**a**) At low speeds. (**b**) At high speeds.

**Figure 9 sensors-25-00874-f009:**
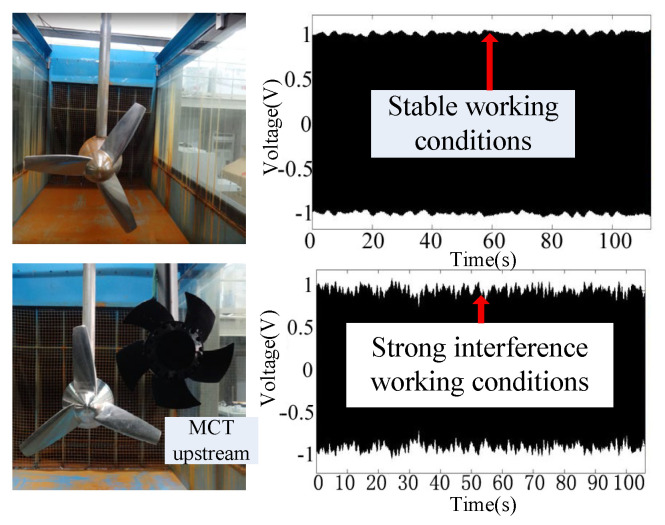
Interference settings.

**Figure 10 sensors-25-00874-f010:**
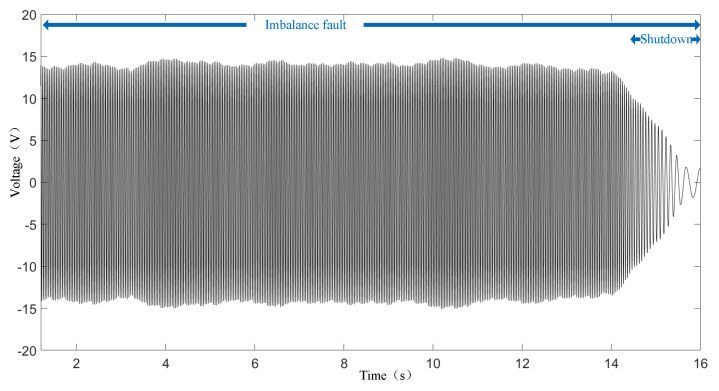
Generator output voltage waveform.

**Figure 11 sensors-25-00874-f011:**
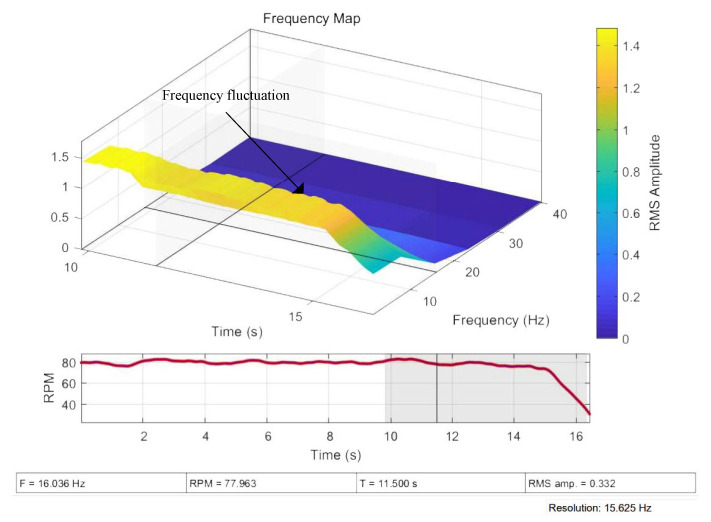
Order analysis of voltage waveform.

**Figure 12 sensors-25-00874-f012:**
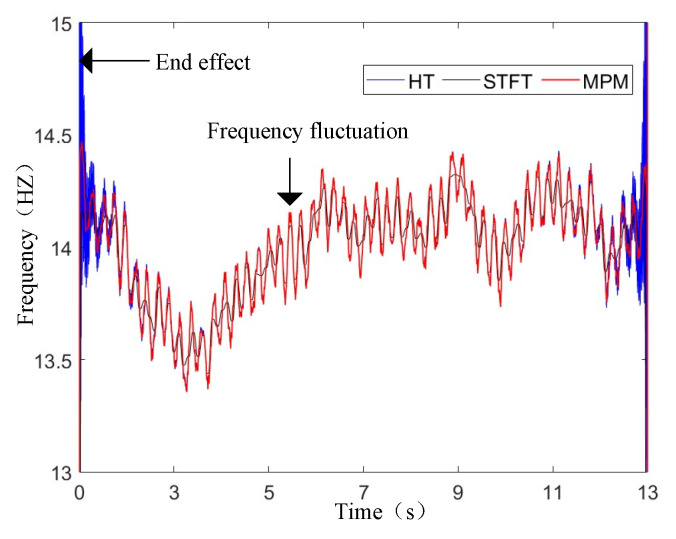
Instantaneous frequency calculated by MPM.

**Figure 13 sensors-25-00874-f013:**
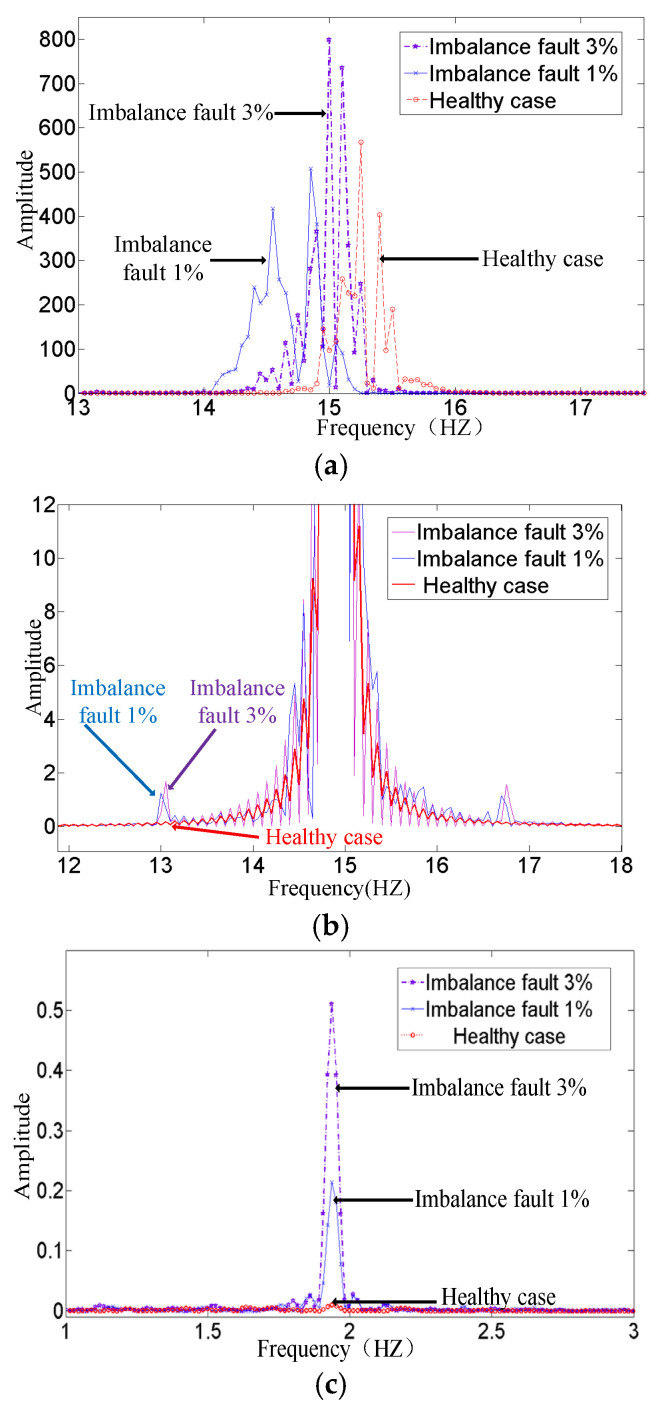
Imbalance detection results with different methods. (**a**) Frequency domain of the voltage. (**b**) Frequency domain after data normalization; (**c**) detection result of proposed method.

**Table 1 sensors-25-00874-t001:** System parameter list.

Turbine	Parameter	Generator	Parameter
Airfoil	Naca0018	Pole-pair	8
Pitch angle	3.4–25.2 deg	Flux	0.1775 Wb
Chord length	5.68–9.6 cm	Resistance	3.3 Ω
Blade diameter	0.6 m	d axis inductance	11.873 mH
Water density	1024 kg/m^3^	q axis inductance	11.873 mH
Water velocity	1.1 m/s	Total inertia	3.5 kg m^2^

**Table 2 sensors-25-00874-t002:** Impeller parameters.

radial position (r/$R_{a}$)	0.178	0.292	0.405	0.518	0.632	0.745	0.858	0.972
Pitch Angle (deg)	25.23	17.83	13.70	10.85	8.61	6.71	5.02	3.47
Chord length (cm)	9.60	9.50	9.24	8.84	8.28	7.56	6.70	5.68

**Table 3 sensors-25-00874-t003:** Comparison of the time-frequency analysis methods.

Fault Degree	Average Value B (Measured Under Stable Flow Conditions)	Relative Error of MPM	Relative Error of STFT	Relative Error of HT
0%	0.007	0.211%	1.032%	1.725%
1%	0.278	0.173%	1.218%	1.531%
3%	0.837	0.165%	4.735%	1.592%

**Table 4 sensors-25-00874-t004:** Detection result yielded by the proposed strategy with different fault degrees.

Fault Degree	Amplitude of the Fault Component	Average Current Frequency	Load Resistance
0%	0.012	15.56 Hz	31.5 Ω
1%	0.104	15.72 Hz	31.5 Ω
3%	0.518	15.36 Hz	31.5 Ω

## Data Availability

The raw data supporting the conclusions of this article will be made available by the authors on request.

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
