# Peer review of "Imbalance Fault Detection of Marine Current Turbine Based on GLRT Detector"

_sensors, 2025, doi:10.3390/s25030874_

Round 1
Reviewer 1 Report
Comments and Suggestions for Authors
This paper presents an imbalance fault detection method for marine current turbines. The proposed approach combines matrix pencil method (MPM), generalized likelihood ratio test (GLRT) and data normalization. Although the described method and the obtained results are of interest, the paper requires improvements.
Major points.
1) The advantages of the proposed approach compared to other available methods should be better emphasized in the introduction. In particular, the differences w.r.t the papers 3. And 7., in which some of the authors were involved, should be highlighted.
2) The term “data normalization” is not used in the introduction, where the authors discuss order spectrum analysis and synchronous sampling. Since data normalization is an important part of the method, it should be mentioned in the introduction and linked to synchronous sampling.
3) The method consists in three main steps: MPM, GLRT, data normalization. The role of each step should be better explained and it is also important to specify what are the inputs and the outputs of each stage. A block diagram of the whole procedure should also be included.
4) The described approach seems very similar to the one reported in the paper
Zhang M , Wang T , Tang T ,et al. An imbalance fault detection method based on data normalization and EMD for marine 357 current turbines[J].ISA Transactions, 2017, vol.68, pp. 302-312.
In particular, the two approaches share the GLRT and the data normalization steps. A comparative analysis of the proposed method with that of the referenced paper should be included in paper, also taking also into account that the experimental setup used is the same in both papers.
5) Many figures should be enlarged to improve readability, in particular Figures 2, 4 and 11.
Other comments.
- Abstract: write “generalized likelihood ratio test” and “matrix pencil method” followed by abbreviations in brackets, as they are used for the first time.
- Figures and figure captions must be in the same page. This is not the case for Figures 2, 4 and 12.
- Figure 4 consists of three subfigures so it is better to use (a), (b) and (c) and explain all subfigures in the caption.
- Line 111: $r_u$ is not a moment, it is a distance.
- In Subsection 3.1 please insert a reference where the MPM is explained.
- The number of complex sinusoids M is estimated through (15) and the selected threshold value is 0.995. Is there a specific reason for this choice? Please explain or provide a proper reference.
- Lines 382-385: two references are merged.
Reviewer 2 Report
Comments and Suggestions for Authors
The paper proposed a fault detection strategy based on GLRT detector. It is very meaningful to accomplish its fault detection to maintain the working order of the MCT system. However, the following should be improved.
1. Abbreviations need to be defined when they first appear. e.g. GLRT, MCT, MPM.
2. Page 1 line 11, there are excess spaces.
3. There should be a space between the sentence and the sequence number. e.g. electricity[3], faults[6], crucial[7].
4. The formula needs to cite references.
5. Page 2 line 85,” Where” >>” where”.
6. The formula needs to be centered, e.g. equation (1), equation (2), equation (3).
7. The title and image of Figure 2 should be on one page.
8. The coordinate axis needs to have units in Figure 2 and 12.
9. No limitation is discussed.
10. The format of references needs to be majorly revised according to Sensors.
9. It is suggested to add a more extended literature review. This literature can strengthen the significance of the paper in academia and industrial worlds. Focus should be given in providing an adequate review in the following topics:
1) Fault diagnosis in rotating machines based on transfer learning: Literature review.
2) Fault diagnosis of rolling bearing based on improved VMD and KNN.
3) A trackable multi-domain collaborative generative adversarial network for rotating machinery fault diagnosis.
4) RTSMFFDE-HKRR: a fault diagnosis method for train bearing in noise environment.
The authors need add the experiment of the detector without GLRT, and compare it with the detector based on GLRT. The conclusion section requires rewriting. Instead of summarizing the methodology, it should critically reflect on the outcomes of the study and their implications. The authors should clearly articulate how the proposed methodology advances the field and acknowledge its limitations.
Round 2
Reviewer 1 Report
Comments and Suggestions for Authors
The authors have addressed the reviewer’s comments.
Here are some minor points:
1) P. 1, line 14: remove ‘method’ after (MPM)
2) P. 3, line 80: insert a space in ‘ofWave’
3) P. 3, line 106: capital letter in ‘a simulation’
4) P.7, line 196: $\theta$ appears as an index
5) P 7, line 197: remove ‘to’
6) P.9, line 239: ‘Table I’ should be ‘Table 1’
7) P. 10, Figure 8: I suggest adding letters (a) and (b) below the subfigures and explaining the meaning of the two subfigures in the caption.
Reviewer 2 Report
Comments and Suggestions for Authors The manuscript has been sufficiently improved to warrant publication in Sensors
Author Response
Thank you again for your advice and help, your suggestion is very pertinent and very helpful for revising and improving our paper.